# KD-PatchMatch: A Self-Supervised Training Learning-Based PatchMatch

**Qingyu Tan, Zhijun Fang * and Xiaoyan Jiang**

School of Electronic and Electrical Engineering, Shanghai University of Engineering Science, Shanghai 201620, China
* Correspondence: zjfang@sues.edu.cn

**Abstract:** Traditional learning-based multi-view stereo (MVS) methods usually need to find the correct depth value from a large number of depth candidates, which leads to huge memory consumption and slow inference. To address these problems, we propose a probabilistic depth sampling in the learning-based PatchMatch framework, i.e., sampling a small number of depth candidates from a single-view probability distribution, which achieves the purpose of saving computational resources. Furthermore, to overcome the difficulty of obtaining ground-truth depth for outdoor large-scale scenes, we also propose a self-supervised training pipeline based on knowledge distillation, which involves self-supervised teacher training and student training based on knowledge distillation. Extensive experiments show that our approach outperforms other recent learning-based MVS methods on DTU, Tanks and Temples, and ETH3D datasets.

**Keywords:** multi-view stereo; learning-based PatchMatch; probabilistic depth sampling; knowledge distillation



## 1. Introduction

Given multiple RGB images with known camera poses, multi-view stereo (MVS) intends to reconstruct a 3D dense point cloud of the image scene. Multi-view stereo has a wide range of applications, including mapping [1], self-driving cars [2], infrastructure inspection [3], robotics [4], etc.

Convolutional neural networks have demonstrated very powerful capabilities in multi-view 3D reconstruction problems in recent years, owing to the continuing development of deep learning. Many learning-based methods [5–8] can incorporate global semantic information, such as specular prior and reflection prior, to improve the robustness of the matching and thus solve the challenges that cannot be overcome by traditional methods. However, MVS still has many challenges, such as untextured areas, occlusion, and non-Lambertian surfaces [9–11].

When MVSNet [7] is proposed, the learning-based MVS domain constructs the cost volume of image pairs using front-to-parallel and differentiable homography. Many subsequent networks are improved on this basis. For example, R-MVSNet [8] innovates the regularization of the cost volume in the depth dimension by using Conv-GRU layer-by-layer processing to reduce the memory consumption; CasMVSNet [12] proposes the first coarse-to-fine structure paradigm to optimize the memory consumption and computational efficiency; Vis-MVSNet [13] and CVP-MVSNet [14] consider in depth the aggregation approach of cost volume and the range of depth assumptions in the subsequent stages of coarse-to-fine from multiple views, respectively, resulting in substantial performance improvements. PatchMatchNet [15] is the first model that introduces the traditional stereo matching algorithm (PatchMatch) into an end-to-end MVS framework.

Most learning-based MVS methods [5,7,8,13,16] employ the same set of depth hypothesis candidates for all pixels (i.e., sampled between hand-picked limits $d_{min}$ and $d_{max}$),

and even the coarse-to-fine approaches [12,14,15,17,18] employ random sampling of candidates to obtain the initial depth map. This has several limitations: in order to find the correct depth, the network evaluates a large number of depth candidates, which wastes a large number of computational resources; in the presence of occlusions, the multi-view consistency assumption is violated; and finally, the reconstruction results become unreliable for untextured areas or reflective surfaces. We believe that the single-view approach can provide a good understanding of the depth of weakly textured or reflective surfaces with proper supervision. Moreover, selecting candidate depths near the single-view depth can improve the efficiency of multi-view matching.

To this end, we propose a new depth sampling method that estimates the single-view probability distribution of each image by the designed network and samples depth candidates in the depth probability distribution estimated from each pixel of the reference image. By sampling depth candidates around the depth of a single view of each pixel, we can improve accuracy while evaluating fewer candidates; With proper supervision, single-view depth estimation [19–21] can learn the depth information of weakly textured regions or reflective surfaces well, so we consider introducing single-view depth estimation in the learning-based PatchMatch as a way to compensate for each other's limitations.

Despite the significant progress of supervised MVS [5,7,8,12–18] in terms of reconstruction quality, ground-truth depth acquisition faces significant challenges for outdoor large-scale scenes. To this end, we introduce a knowledge distillation-based self-supervised training pipeline for the PatchMatch-based networks, which consists of two main steps: the generation of pseudo-probabilistic knowledge and student training. The teacher model is made to train in a self-supervised manner by introducing feature-metric loss in the teacher model, and the pseudo-probability of the teacher model's distribution is generated by probabilistic coding. It is possible to transfer the teacher model's probabilistic knowledge to the student model by forcing the student model to have a predicted probability distribution similar to the pseudo-probability distribution. After our experiments on different datasets, the performance of the student model outperforms the teacher model.

In summary, our main contribution is a new PatchMatch-based MVS framework, which combines the advantages of single-view depth estimation with probabilistic depth sampling in the depth initialization phase and introduces a self-supervised training pipeline based on knowledge distillation, requiring multiple innovations:

- We introduce a single-view depth estimation network in the PatchMatch-based MVS framework, where higher accuracy can be obtained while evaluating fewer candidates by using probabilistic depth sampling in the depth initialization phase.
- We introduce a knowledge distillation-based PatchMatch self-supervised training pipeline that enables the network to be trained in a self-supervised manner and to transfer the generated probabilistic knowledge to the student model.
- The sensitivity of photometric loss to shooting angles and lighting conditions leads to poorer completeness of model predictions. To better train the teacher model, we add the internal feature metric consistency loss to the original photometric loss, i.e., add the photometric loss computed between internal feature maps, allowing robust self-supervised training of the teacher model.

In this paper, we introduce related works on multi-view stereo and its sampling methods in Section 2. We improve learning-based PatchMatch by introducing a new sampling method and propose a training pipeline based on knowledge distillation in Section 3. We demonstrate the effectiveness and generalization of our method by conducting extensive experiments on three different datasets in Section 4.

## 2. Related Works

**Geometry-Based Multi-View Stereo.** Multi-view stereo (MVS) is a process of generating dense correspondences from multiple images of known camera poses to produce dense 3D point cloud reconstruction results. According to the output result representation, traditional MVS methods are classified into three categories: voxel-based reconstruction [19,20],

point-cloud-based reconstruction [21,22], and depth-map-based reconstruction [23–25]. In contrast, the depth-map-based methods are more efficient. This class of methods is generally based on the iterative optimization of PatchMatch [23–25] to estimate the depth map. Galliani et al. [23] propose Gipuma, which adopts a red–black checkerboard grid propagation approach that makes full use of GPUs to achieve massively parallel operations and greatly improves the efficiency of the algorithm. Schönberger et al. [24] propose COLMAP, which utilizes a more complex view selection strategy based on the Markov chain model. ACMM [25] proposes a multiscale geometric consistency bootstrap framework that can be used at different scales to sense remarkable information and to convey information through the geometric consistency of multiple views. Although these methods have achieved relatively good results, they still fail to match for untextured areas or non-Lambertian surfaces.

**Learning-Based PatchMatch.** To handle more complex real-world scenes, particularly without textured or reflective surfaces, recent learning-based MVS methods [2,15,17] use CNNs to extract robust features and perform matching, as well as post-processing by implementing the entire traditional stereo pipeline as a neural network layer, allowing the model to be trained end-to-end, greatly improving efficiency. Duggal et al. [2] proposed a differentiable PatchMatch module to predict the parallax range per pixel and construct a more efficient sparse cost quantity. Galliani et al. [15] proposed PatchMatchNet, a novel and learnable cascade formulation, which improves the PatchMatch core algorithm by introducing an adaptive approach that causes a significant drop in computation time and memory consumption. PatchMatch-RL [17] uses PatchMatch optimization to estimate pixel depth, normals, and visibility while minimizing the expected photometric cost and maximizing the likelihood of ground-truth depth and normals by using reinforcement learning. Similarly, we introduce single-view depth estimation into the PatchMatch initialization part, which enables the model to evaluate fewer depth candidates while achieving higher accuracy and combine with knowledge distillation to propose a self-supervised training pipeline that overcomes the difficulty of ground-truth depth acquisition.

**Coarse-to-Fine Depth Sampling.** Most learning-based MVS methods [12,13,15,18,19] use the same depth hypothesis candidate set for all pixels, and to obtain higher accuracy, the depth candidates should be densely sampled, but this leads to huge memory consumption and long computation time. Recent MVS approaches [12,14,18] use a coarse-to-fine strategy to construct multi-scale cost volumes to get around this constraint, abandoning the previous construction of cost volumes at a fixed resolution, which results in lower computation time and GPU memory consumption. Ref. [12] obtains the depth ranges of different resolution layers and their intervals according to a fixed formula. Ref. [18] obtains the depth hypotheses of different resolution layers by acquiring the points of neighboring pixel projections of the source image. Ref. [14] introduces an adaptive resolution cost-volume-based depth estimation method, where the depth hypotheses space of each layer is obtained from the uncertainty of the pixel projections of the previous layer. We introduce probabilistic depth sampling where the depth candidates for each pixel are sampled from the single-view depth probability distribution of each pixel, which allows for higher accuracy while evaluating fewer depth candidates.

**Knowledge Distillation.** Knowledge distillation [26] is a method of extracting knowledge from cumbersome models and compressing it into a single model so that it can be deployed into real-world applications. A small "student" model in knowledge distillation learns to copy a large "teacher" model and utilizes the teacher's expertise to achieve parity or greater accuracy. The three main types of methods that are used to train student and teacher models are offline, online, and self-distillation. The most widely known approach that employs a trained instructor model to direct the student model is offline distillation [27,28]. In this approach, the teacher model is first pre-trained on the training dataset, and then the knowledge extracted from it is used to train the student model. PKT [29] matches probability distributions of data in the feature space formed by the teacher and student models rather than just regressing their actual representations,

such that learning through probability distributions preserves teachers' quadratic mutual information (*QMI*) [30] in smaller student models.

## 3. Method

In this section, we introduce the framework of KD-PatchMatch, illustrated in Figure 1. We describe the part of probabilistic sampling in Section 3.1, where we focus on how to sample per-pixel depth candidates during initialization. Finally, we introduce a self-supervised training pipeline for our network based on knowledge distillation in Section 3.2, which consists of teacher training (Section 3.2.1) and distillation-based student training (Section 3.2.2).

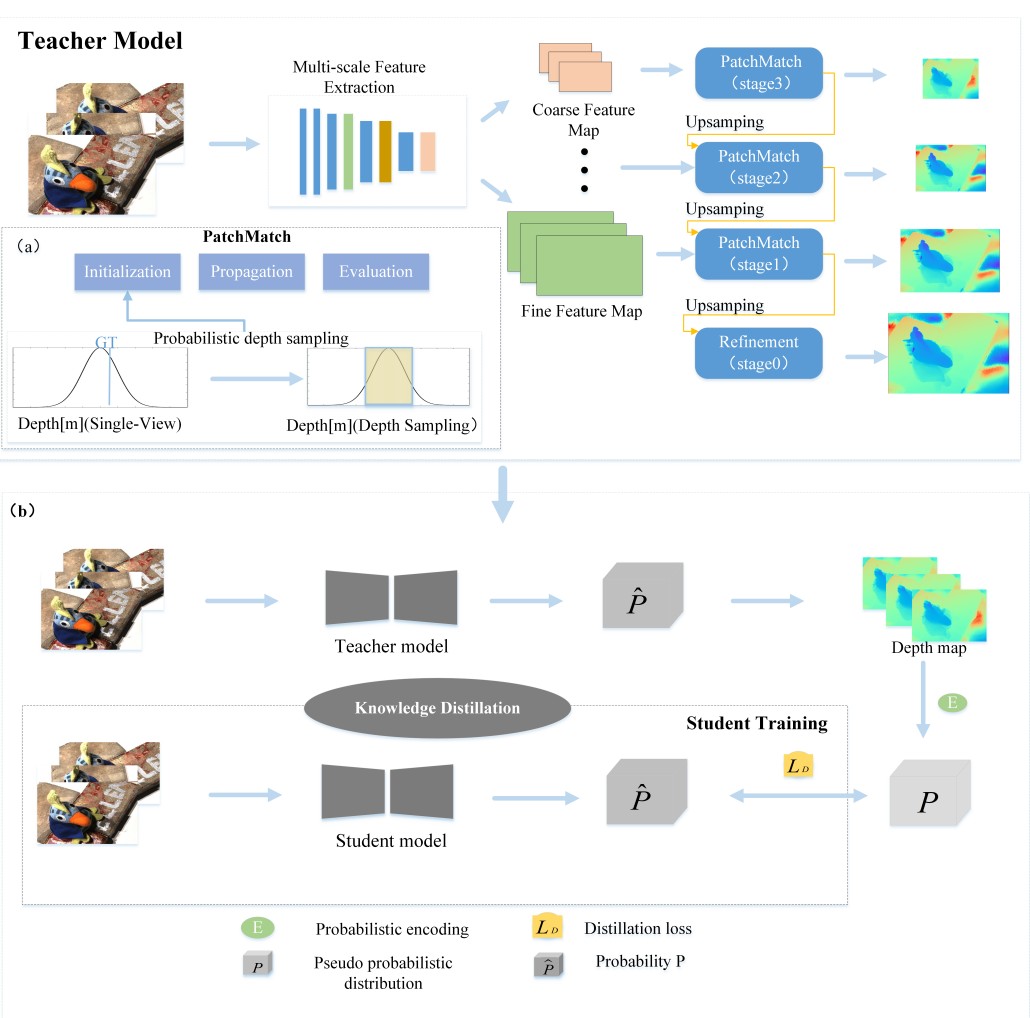

**Figure 1.** The framework of KD-PatchMatch, including PatchMatch-based MVS framework and self-supervised training pipeline based on knowledge distillation: (**a**) This figure illustrates how the depth candidates are sampled during the depth initialization phase, The curves and shadows represent the estimated depth probability distributions and depth sampling range during the depth initialization phase. (**b**) The pipeline for self-supervised training based on knowledge distillation is shown in this image, along with the creation of pseudo-probabilistic knowledge and student training.

### 3.1. Probabilistic Depth Sampling

With $N$ input images of size $W \times H$, including a reference image $I_0$ and its neighboring source images $\{I_i\}_{i=1}^{N-1}$, in the coarse-to-fine optimization, features are extracted hierarchically at different resolution levels by using a Feature Pyramid Network (FPN) [31]. At Stage $k$, the resolution of feature maps is $\frac{W}{2^k} \times \frac{H}{2^k}$.

**Single-View Depth Probability.** At Stage 3 of PatchMatch, the depth initialization is originally performed in a random way and uses the same set of depth candidates for all pixels. In order to find the correct depth, the network has to evaluate a large number of depth candidates, resulting in a large number of computational resources being wasted. To this end, following Gwangbin Bae et al., 2022 ([32]), we propose a depth sampling method based on D-Net [32], which allows the network to perform better while considering fewer depth candidates. Instead of uniform sampling in image space, we sample per-pixel depth candidates for input images by the single-view depth probability distribution. For each image of size $W \times H$, our network estimates a single-view depth probability distribution map at the resolution $\frac{H}{8} \times \frac{W}{8}$ of the input image $I_0$. The depth distribution of each pixel $p_0$ in the reference image $I_0$, $P_{p_0}(d \mid I_0)$ is parameterized as a Gaussian distribution,

$$P_{p_0}(d \mid I_0) = \frac{1}{\sigma_{p_0}(I_0)\sqrt{2\pi}} e^{-\frac{1}{2}\left(\frac{d - \mu_{p_0}(I_0)}{\sigma_{p_0}(I_0)}\right)^2} \tag{1}$$

where $d$ is the depth value of pixel $p_0$, and $\mu_{p_0}$ and $\sigma_{p_0}$ are the mean and the variance of depth values of pixel $p_0$. Efficient-Net B5 [33] is used as a lightweight convolutional network encoder–decoder. Our single-view network is pre-trained, with the weights fixed while the rest of the pipeline is being trained. The training loss is defined as negative log-likelihood (*NLL*) of the ground-truth depth,

$$L_{p_0}\left(d_{p_0}^{gt} \mid I_0\right) = \frac{1}{2}\log \sigma_{p_0}^2(I_0) + \frac{\left(d_{p_0}^{gt} - \mu_{p_0}(I_0)\right)^2}{2\sigma_{p_0}^2(I_0)} \tag{2}$$

Equation (2) is an *L2* loss with a decaying learning rate; $d_{p_0}^{gt}$ is the ground truth of the depth value of pixel $p_0$. When reducing the error $((d^{gt} - \mu)^2$ is difficult, the $\sigma^2$ estimated by network learning will be high. This usually happens at points near and far from the object boundary [34]. Conversely, when the estimated $\sigma^2$ is low, the correct depth may be near the estimated $\mu$.

**Probabilistic Depth Sampling.** We use the single-view depth probability distribution estimated for the reference image $I_0$ for the pixel-by-pixel depth hypothesis sampling. First, we need to define the search space $[\mu_{p_0} - \beta\sigma_{p_0}, \mu_{p_0} + \beta\sigma_{p_0}]$ for each pixel $p_0$, where $\beta$ is the hyperparameter, and we will assign a value to $\beta$ in the experimental part. Then, we divide the search space into $D_f$ intervals, ensuring that each interval has the same probability mass. Finally, the midpoint of each interval is chosen as the depth candidate. This is performed so that more candidates are sampled around $\mu_{p_0}$ (i.e., the most probable depth value). The *j*-th depth candidate hypothesis, $d_{p_0}^j$, is defined as

$$d_{p_0}^j = \mu_{p_0} + b_j \sigma_{p_0} \tag{3}$$

where

$$b_j = \frac{1}{2}\left[\Phi^{-1}\left(\frac{j-1}{D_f}P^* + \frac{1-P^*}{2}\right) + \Phi^{-1}\left(\frac{j}{D_f}P^* + \frac{1-P^*}{2}\right)\right]. \tag{4}$$

In Equation (4), $\Phi^{-1}()$ is the probit function, and $P^* = erf(\frac{\beta}{\sqrt{2}})$ is the probability mass covered by the interval $[\mu_{p_0} \pm \beta\sigma_{p_0}]$. Note that the value of $b_j$ depends only on $D_f$ and $\beta$, independent of the pixel points themselves. Figure 2 demonstrates uniform sampling versus probability sampling. We can improve accuracy while evaluating fewer candidates because we only sample deep candidates within the $\beta$-sigma confidence interval.

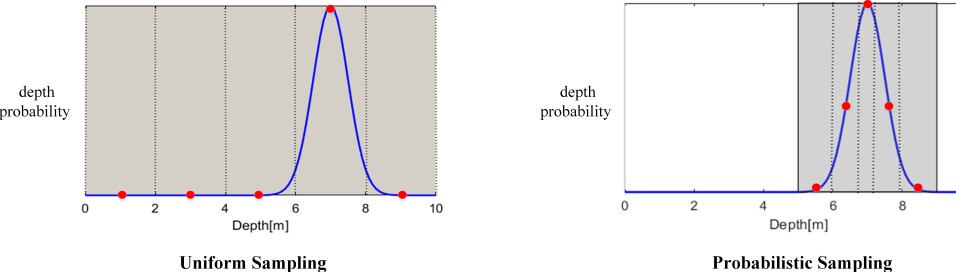

**Figure 2.** Comparison of uniform and proposed probabilistic sampling methods. The single-view depth probability distribution is represented by the blue curve, and the sampled candidates are represented by the red dots.

### 3.2. Knowledge Distillation

Despite the significant progress of supervised MVS [12–15] in terms of reconstruction quality, ground-truth depth acquisition faces significant challenges for outdoor large-scale scenes. To this end, based on KD-MVS [35], we propose a self-supervised training pipeline, which transfers the knowledge from a large to a small model without sacrificing validity. Following Yikang Ding et al., 2020 ([35]), our pipeline primarily comprises teacher training (Section 3.2.1) and distillation-based student training (Section 3.2.2).

#### 3.2.1. Teacher Training

The sensitivity of photometric loss to shooting angles and lighting conditions leads to poorer completeness of model predictions. To this end, in addition to the traditional photometric consistency used in our network, following [35], we propose to add the internal feature metric consistency loss to the original loss as additional supervised signals, which are obtained by computing the luminosity loss between the reference and source views. The view reconstruction and loss functions are described below.

**View Reconstruction.** For a pixel $p_0$ in the reference image $I_0$, given the depth value of the $j$-th sample, $d_{p_0}^j$, its corresponding pixel $p_i(d_{p_0}^j)$ in the source image $I_i$ is calculated as

$$p_i\left(d_{p_0}^j\right) = K_i\left(R_{0,i} \cdot \left(K_0^{-1} \cdot p_0 \cdot d_{p_0}^j\right) + t_{0,i}\right) \tag{5}$$

where $R_{0,i}$ and $t_{0,i}$ are the relative rotation and translation of the reference image $I_0$, and the source images $I_i$, $k_0$, and $k_i$ are the intrinsic matrices of the reference image $I_0$ and the source image $I_i$. With the above transformation, the warped feature maps of the source view under the $j$-th set of (per pixel different) depth hypotheses, $F_i(p_i(d_{p_0}^j))$, can be obtained by differentiable bilinear interpolation.

**Loss Function.** The training loss consists of two components: photometric loss $L_{ph}$ and feature-metric loss $L_{fm}$. Relying on photometric loss based on the $L_1$ distance between the original RGB reference image and the reconstructed image, previous work can result in large systematic errors in realistic scenes due to reflective surfaces and occlusions. To end this, following [36], we add the feature-metric loss to the photometric loss, i.e., adding the calculation of the photometric loss between feature maps. Feature-metric loss is defined as

$$L_{fm} = |F_i(\hat{p}) - F_0(p)|_1 \tag{6}$$

where $F_i(\hat{p})$ and $F_0(p)$ represent the reference view's feature map and the reconstructed feature map from the $i$-th source view, respectively.

In the refinement stage, we upsample (from resolution $\frac{W}{2^k} \times \frac{H}{2^k}$ to $W \times H$) the finest resolution level (Stage 0) through the designed depth residual network and obtain the photometric loss compared to the ground truth, i.e., $L_{ref}^0$.

To summarize, the final loss function includes all the depth estimation, and the refinement stage is defined as

$$L_{\text{totol}} = \sum_{k=1}^{3} \sum_{m=1}^{n_k} \left( \lambda_{fm} L_{fm}^{m,k} + \lambda_{ph} L_{ph}^{m,k} \right) + L_{ref}^0 \tag{7}$$

where $L_{fm}^{m,k}$ and $L_{ph}^{m,k}$ are the feature-metric loss and photometric loss of the $m$-th iteration of PatchMatch at Stage $k(k = 1, 2, 3)$, and $L_{ref}^0$ is the loss of final depth map refinement.

### 3.2.2. Distillation-Based Student Training

In this section, we train our network using the knowledge distillation concept, which entails transferring the teacher's probabilistic knowledge to the student model. This procedure is divided into two parts: pseudo-probabilistic knowledge generation and student training.

**Pseudo Probabilistic Knowledge Generation.** Similar to [37], knowledge is transferred using the probability distribution. However, when applying knowledge distillation in the PatchMatch-based MVS network, we discover that the teacher model's true probabilistic knowledge is not directly available for training the student model. The reason for this is that the depth hypotheses in the coarse-to-fine network must be dynamically sampled based on the previous stage's results, i.e., in subsequent iterations of the PatchMatch stage, we need to use the estimates from the previous iteration, possibly sampled from the previous coarse stage, so that the teacher and student models cannot always share the same depth hypotheses. To this end, following [35], we use probabilistic coding to generate pseudo-probability distributions.

We denote the final depth prediction as $D_0$, i.e., the depth map generated after the refinement, the depth maps of source views as $\{D_i\}_{i=1}^{N-1}$. We cast the 2D point $p_0$ into 3D space for any given pixel coordinate in the reference image to obtain the 3D point with the depth value $D_0(p_0)$. Then, to obtain the point $p_i$ in the source view, we back project $P_0$ to the $i$-th source view. After that, using the estimated depth $D_i(p_i)$ in the reference image, we cast $p_i$ into the 3D space to obtain the 3D point $P_i$. Finally, the depth value of $P_i$ observed from the reference camera is denoted as $D_0(P_i)$.

**Probabilistic Encoding.** We use the $\{D_0(P_i)\}_{i=0}^{N-1}$ to generate the pseudo probability distribution $P_{p_0}(d)$ for the depth value $d$ of each pixel $p_0$ in the reference view. By using maximum likelihood estimation (*MLE*), we model it as a Gaussian distribution with mean depth value $\mu_{p_0}$ and variance $\sigma_{p_0}^2$:

$$\mu_{p_0} = \frac{1}{N} \sum_{i=0}^{N-1} D_0(P_i) \tag{8}$$

$$\sigma_{p_0}^2 = \frac{1}{N} \sum_{i=0}^{N-1} (D_0(P_i) - \mu_{p_0}) \tag{9}$$

$\mu_{p_0}$ fuses the depth information of the reference and source images, while $\sigma_{p_0}^2$ reflects the uncertainty of the teacher model's depth value at pixel $p_0$. During the distillation training, it will provide probabilistic knowledge to the student model.

**Student Training.** With the above pseudo probability distribution $P$, by making the projected probability distribution of a student model, $P$, resemble the pseudo-probability distribution, $\hat{P}$, one can train the model from scratch. For the discrete depth hypothesis $\{d_j\}_{j=1}^{D_f}$, we compute their pseudo-probabilities $\{P(d_j)\}_{j=1}^{D_f}$ over the continuous probability distribution $P$ and normalize $\{P(d_j)\}_{j=1}^{D_f}$ with SoftMax, resulting in the final discrete pseudo-probability value. The Kullback–Leibler divergence is used to calculate the

difference between the student model's predicted and pseudo-probabilities. The distillation loss $L_{D_f}$ is defined as

$$L_{D_f} = L_{KL}(P\|\hat{P}) = \sum_{p\in\{p_0\}} (P_p - \hat{P}_p) \log\left(\frac{P_p}{\hat{P}_p}\right) \tag{10}$$

where $\{p_0\}$ represents the pixel in the reference image $I_0$.

## 4. Experiments

### 4.1. Datasets

The DTU dataset [9] is an indoor dataset specifically captured and processed for MVS, including 124 different objects or scenes. We divide the dataset into training scans, testing scans, and validation scans by following [38]. The Tanks and Temples dataset [10] is a public benchmark obtained under realistic conditions, which contains intermediate subsets of eight scenes and advanced subsets of six scenes. The ETH3D benchmark [11] contains calibrated high-resolution scene images with strong viewpoint variation, which is split into training and test datasets.

### 4.2. Implementation Details

We use PyTorch to implement the model and train it on the DTU training set [9]. The number of input images is set to $N = 5$, and the image resolution is set to $640 \times 512$. The selection of source images and the iteration number of PatchMatch at Stage $3, 2, 1$ are consistent with PatchMatchNet [15]. For probabilistic depth sampling, we set $D_f = 8$, $\beta = 3$. For loss function, we set $\lambda_{fm}$ and $\lambda_{ph}$ as four and one, respectively. The teacher model is trained with Adam for eight epochs, setting the learning rate to 0.001. In the student training phase based on the distillation method, the student model was trained with a pseudo-probability distribution for 12 epochs. Here, we train on 1 Nvidia GTX 1080Ti GPU. Similar to MVSNet [7], we reconstruct point clouds after depth estimation.

### 4.3. Experimental Results

**Evaluation on DTU Dataset.** We set $N = 5$ and input images resolution as $1600 \times 1200$. We use the DTU dataset's [9] three evaluation metrics, accuracy, completeness, and overall. As shown in Table 1, our method outperforms others in terms of completeness and overall quality, despite the fact that Gipuma [23] performs best in terms of accuracy. Figure 3 depicts a comparison of reconstructed point clouds. This result shows that our method achieves significantly higher reconstruction quality than PatchMatchNet [15] and other supervised methods, such as the roof reconstruction is more complete and provides more precise borders, and the logo letters have finer details.

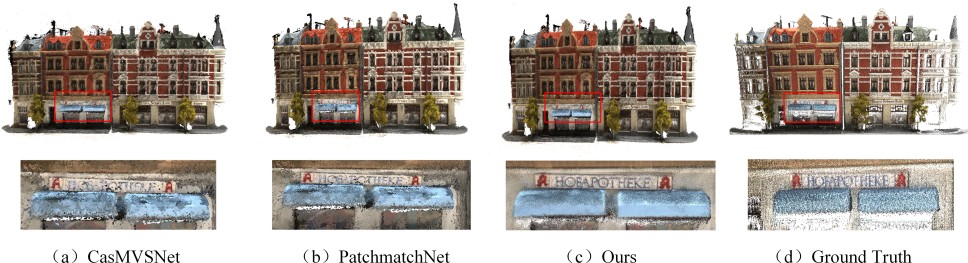

| （a）CasMVSNet | （b）PatchmatchNet | （c）Ours | （d）Ground Truth |

**Figure 3.** Qualitative comparion of scan15 of DTU [9]: bottom, zoom in; our method preserves the integrity of the blue roof better than PatchMatchNet [15], with smoother results and finer detail in the logo letters.

Since our method samples depth values for each pixel around the single-view probability distribution, multi-view matching may fail for texture-free surfaces. In contrast to

single-view which is more reliable, our method is 6.8% more effective in terms of completeness compared to PatchMatchNet [15]. In addition, our method introduces a self-supervised pipeline based on knowledge distillation, which transfers the deep pseudo-probability knowledge from the teacher model to the student model. After our experiments, the student model is better than the teacher model. Our method is 8.2% more effective in terms of accuracy compared to PatchMatchNet [15]. Combining the results in Table 1, the overall reconstruction effect of our method is 7.7% better than PatchMatchNet [15]. We visualize reconstructed point clouds from DTU's evaluation set [9] in Figure 4.

**Table 1.** Quantitative results of different methods on DTU's evaluation set [9] (lower is better).

| Methods | Acc. (mm) | Comp. (mm) | Overall (mm) |
| --- | --- | --- | --- |
| Camp [39] | 0.835 | 0.554 | 0.695 |
| Furu [21] | 0.613 | 0.941 | 0.777 |
| Tola [40] | 0.342 | 1.190 | 0.766 |
| Gipuma [23] | **0.283** | 0.873 | 0.578 |
| SurfaceNet [38] | 0.450 | 1.040 | 0.745 |
| MVSNet [7] | 0.396 | 0.527 | 0.462 |
| R-MVSNet [8] | 0.383 | 0.452 | 0.417 |
| CIDER [6] | 0.417 | 0.437 | 0.427 |
| P-MVSNet [5] | 0.406 | 0.434 | 0.420 |
| Point-MVSNet [16] | 0.342 | 0.411 | 0.376 |
| Fast-MVSNet [41] | 0.336 | 0.403 | 0.370 |
| CasMVSNet [12] | 0.325 | 0.385 | 0.355 |
| UCS-Net [18] | 0.338 | 0.349 | 0.344 |
| CVP-MVSNet [14] | 0.296 | 0.406 | 0.351 |
| PatchMatchNet [15] | 0.427 | 0.277 | 0.352 |
| ours | 0.392 | **0.258** | **0.325** |

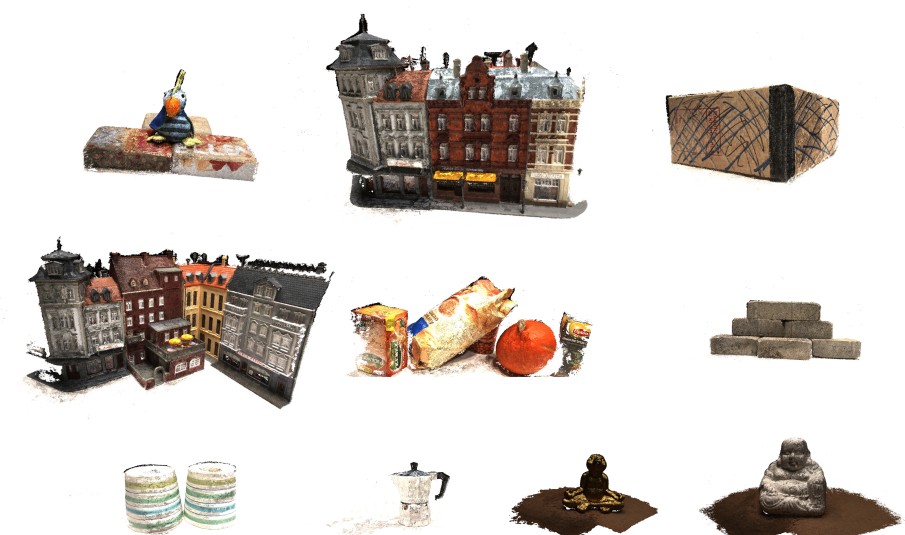

**Figure 4.** Reconstruction results on a partial evaluation set of DTU [9].

**Evaluation on Tanks and Temples Dataset.** We test our method on the Tanks and Temples benchmark [10] to demonstrate its generalization ability on different data. Without any fine-tuning, we employ the model trained on DTU [9]. In evaluation, we set $N = 7$ and image resolution as $1920 \times 1056$. We recover the camera parameters and sparse point clouds using OpenMVG [42]. On intermediate datasets, our method performs similarly to CasMVSNet [12], as shown in Table 2. For reconstructing more difficult advanced datasets, our method outperforms the best in all of the methods. Overall, our method demonstrates competitive generalization performance. We visualize reconstructed point clouds from the Tanks and Temples dataset [10] in Figure 5.

**Table 2.** Quantitative results of different methods on Tanks and Temples [10] (F score, higher is better). Note that most methods avoid evaluation on more challenging advanced datasets.

| Methods | Intermediate | Advanced |
|---|---|---|
| COLMAP [24] | 42.14 | 27.24 |
| MVSNet [7] | 43.48 | - |
| R-MVSNet [8] | 48.40 | 24.91 |
| CIDER [6] | 46.76 | 23.12 |
| P-MVSNet [5] | 55.62 | - |
| Point-MVSNet [16] | 48.27 | - |
| Fast-MVSNet [41] | 47.39 | - |
| CasMVSNet [12] | **56.42** | 31.12 |
| UCS-Net [18] | 54.83 | - |
| CVP-MVSNet [14] | 54.03 | - |
| PatchMatchNet [15] | 53.15 | 32.31 |
| ours | 56.36 | **33.68** |

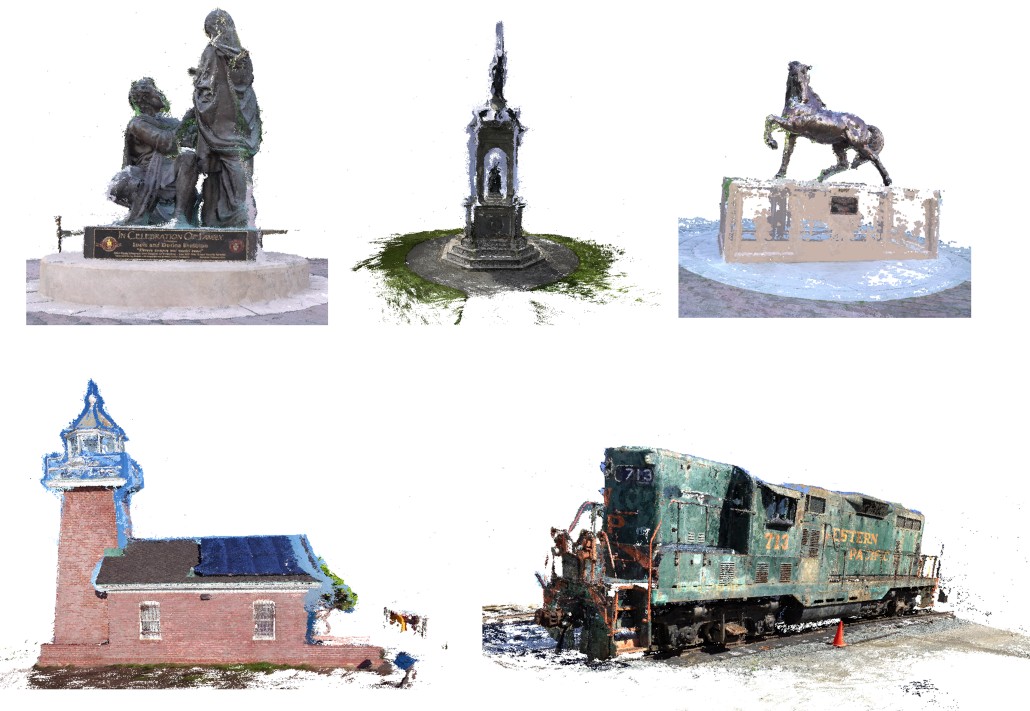

**Figure 5.** Reconstruction results on the Tanks and Temples dataset [10].

**Evaluation on ETH3D Benchmark.** Without any fine-tuning, we employ the model trained on DTU [9]. For evaluation, we set the size of the input image to 2688 × 1792. Because of the strong viewpoint variation in ETH3D [11], we also use a view with $N = 7$ to exploit more multi-view information. We recover camera parameters and sparse point clouds using COLMAP [24]. Most learning-based MVS methods cannot produce reasonable results on the ETH3D dataset [11] due to viewpoints, but our approach can cover weakly textured areas, including white walls and columns, while keeping high accuracy.

Due to the probability distribution sampling, we drastically reduce the number of depth hypothesis samples (i.e., eight candidates are used). As shown in Table 3, compared to COLMAP [24] and PVSNet [43], we improve the accuracy while greatly reducing the computation time on the training dataset. On the more challenging test dataset, the performance of our method is better than PatchMatchNet [15], but the run time is comparable to it. We visualize reconstructed point clouds from the ETH3D benchmark [11] in Figure 6.

**Table 3.** Quantitative results of different methods on ETH3D [11] (F score, higher is better). Due to the high variability of viewpoints, the only competitive purely learning-based approach currently submitted on ETH3D is PVSNet [43].

| Methods | Training | | Test | |
|---|---|---|---|---|
| | **F1 Score** | **Time (s)** | **F1 Score** | **Time (s)** |
| MVE [44] | 20.47 | 13278.69 | 30.37 | 10550.67 |
| Gipuma [23] | 36.38 | 587.77 | 45.18 | 689.75 |
| PMVS [21] | 46.06 | 836.66 | 44.16 | 957.08 |
| COLMAP [24] | 67.66 | 2690.62 | 73.01 | 1658.33 |
| PVSNet [43] | 67.48 | - | 72.08 | 829.56 |
| PatchMatchNet [15] | 64.21 | **452.63** | 73.12 | **492.52** |
| Ours | **67.92** | 637.49 | **74.86** | 639.30 |

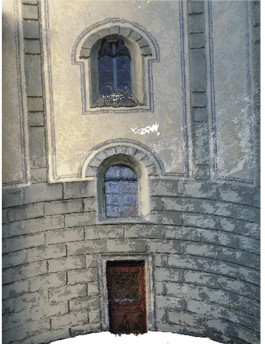 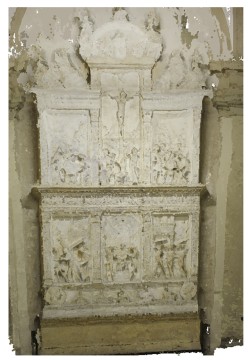 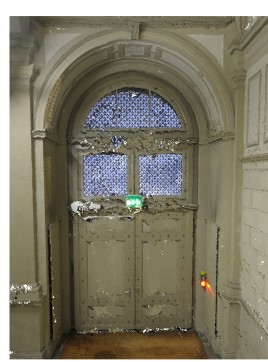

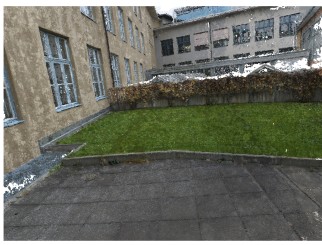 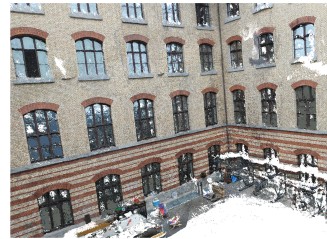 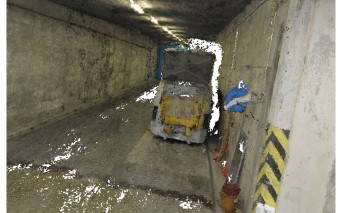

**Figure 6.** Reconstruction results on the ETH3D benchmark [11].

### 4.4. Ablation Study

To evaluate the efficacy of the suggested probabilistic depth sampling and feature-metric loss, we undertake ablation studies in this section. The DTU dataset [9] is used as the basis for all of the research.

**Effectiveness of the Proposed Probabilistic Depth Sampling.** We compare the effects of multi-view reconstruction with and without the proposed probability sampling. Since the inverse depth sampling used by PatchMatchNet [15] is essentially a uniform sampling on the image space, which samples 48 depth candidates in the depth initialization phase, we use probabilistic depth sampling to sample 8 candidates in the depth initialization phase. As shown in Figure 7, our method reconstructs better while sampling fewer candidates.

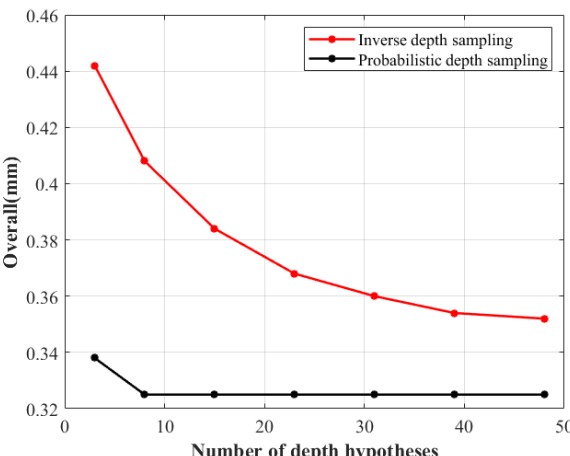

**Figure 7.** Effectiveness of the proposed probabilistic depth sampling. We compared probabilistic depth sampling with the original inverse depth sampling of the teacher model.

**Effectiveness of the Feature-Metric Loss.** It is important to note that using feature-metric loss alone is not feasible. The reason for this is that the feature network in the MVS network is trained online. As a result, using feature metric loss alone will result in training failure when the features are constant. In Table 4, we compare the effects of adding and not adding feature-metric loss in the model.

**Table 4.** Ablation study on feature-metric loss in the teacher model. $L_{pm}$ and $L_{fm}$ denote photometric loss and feature-metric loss respectively.

| $L_{pm}$ | $L_{fm}$ | Acc. (mm) | Comp. (mm) | Overall. (mm) |
|:---:|:---:|:---:|:---:|:---:|
| ✓ | | 0.427 | 0.277 | 0.352 |
| ✓ | ✓ | **0.419** | **0.265** | **0.342** |

## 5. Discussion

### 5.1. Insights of Effectiveness

Compared with traditional deep-learning-based MVS frameworks, our approach effectively saves computational resources and has good generalization properties and competitive performance. We conducted extensive experiments on the DTU, Tanks and Temples, and ETH3D datasets to verify the effectiveness of the method. From Table 3, we can see that our method can effectively save running time compared with the traditional deep-learning-based MVS method, and the F1 score (reconstruction quality) has a significant improvement.

### 5.2. Insights of Self-Supervision

Our self-supervised approach generates deep pseudo-labels through the training of a teacher model. We believe that the pseudo-labels generated by the teacher model screened out most of the wrong samples, and of course, there are a small number of "toxic samples".

Therefore, in future work, we consider adding some deep pseudo-label verification and obtaining more accurate pseudo-labels by setting strict threshold conditions. The verified pseudo-label is better than GT to a certain extent. This is due to the verified pseudo-label being able to filter out some outliers, so that the training will be more stable, and the performance will be improved.

### 5.3. Applicability of the Method

Our self-supervised approach is suitable for some memory-constrained devices or time-critical applications. In addition, for outdoor large-scale datasets, our method shows

good performance and generalization ability, which is due to the fact that large-scale datasets can provide enough data diversity for knowledge distillation, and student models will learn robust feature representations.

## 6. Conclusions

In this paper, we propose KD-PatchMatch, a novel PatchMatch-based MVS framework. Combined with single-view depth estimation, probabilistic depth sampling is used in the initialization part of PatchMatch to sample the depth hypothesis, and the model is trained using a self-supervised training pipeline based on knowledge distillation. Extensive experiments on DTU, Tanks and Temples, and ETH3D show that it has low computation time, good generalization performance, and competitive performance compared to state-of-the-art algorithms, but there is still potential for further improvement of our model. For example, the network architecture is redundant and has limitations for the reconstruction of illumination changes, untextured areas, and non-Lambertian surfaces. We can introduce traditional methods or methods from other fields, such as geometric checks, to make the reconstruction results better.

**Author Contributions:** Conceptualization, Q.T. and Z.F.; methodology, Q.T.; software, Q.T.; validation, Q.T.; formal analysis, Q.T.; investigation, Q.T.; resources, Z.F.; data curation, Q.T.; writing—original draft preparation, Q.T.; writing—review and editing, Z.F. and X.J.; visualization, Q.T.; supervision, Z.F. and X.J.; project administration, Z.F.; funding acquisition, Z.F. All authors have read and agreed to the published version of the manuscript.

**Funding:** This research was supported in part by the National Natural Science Foundation of China under (Grant number U2033218).

**Institutional Review Board Statement:** Not applicable.

**Informed Consent Statement:** Not applicable.

**Data Availability Statement:** Data will be made available on request.

**Conflicts of Interest:** The authors declare no conflict of interest.

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
