# Peer review of "KD-PatchMatch: A Self-Supervised Training Learning-Based PatchMatch"

_applsci, doi:10.3390/app13042224_

Round 1

Reviewer 1 Report

Questions

1. How to check the validity of the proposed KD-patchMatch, a novel patchmatch-based MVS framework?

2. During self supervised proposed method if some errors come, then there is any self alarming for the user to correct the errors?

3. At what instant you can make successful the proposed method in our real life and how to satisfy the proposed method is more efficient likewise cost and time saving than the existing methods?

Suggestion

Please add a section in which above questions and there answers helper the reader to know more about your proposed method.

Reviewer 2 Report

a. The first abbreviation should be explain like MvS etc

b. The main finding is not clear, kindly revise for abstract

c. Please give the main motovation and gap in introduction

d. Please add organized of paper in the end introduction

e. Please check subsection 3.1, I suggest author explain variable after equation

f. Check Figure 2, y-axis is missing

g. This article should be proofread

Reviewer 3 Report

This manuscript proposed the KD-pathMatch, a patchmatch-based multi-view stereo (MVS) framework. There are a few comments as follows:

1. In the framework of KD-patchMatch shown in Figure 1, (a) illustrates how the depth candidates are sampled during the depth initialization phase. There are 3 boxes below PatchMatch, initialization, propagation, and evaluation. Is it possible to give more information about this subfigure (a)?

2. In the coarse-to-fine optimization, features are extracted by feature pyramid network, is the output of this step multi-scale feature extraction in Figure 1? How to determine the number of stages for different problems?

3. For the total loss function given in equation (7), there are 3 terms, loss ref, loss fm, and loss ph, how to determine the weights of different terms? In line 227, the weights are set as 4 and 1, but why? Besides, the loss function is also related to the number of stages, how to adjust the loss weights if the number of stages varies?

4. In addition to the weights in loss function, there are other hyperparameters in the network, like Df and \beta in probabilistic depth sampling. How to determine those hyper-parameters? A detailed discussion on the determination of all hyper-parameters is needed for real application. 

Round 2

Reviewer 2 Report

This form can be accept